# A Conceptual Model for Development of Small Farm Management Information System: A Case of Indonesian Smallholder Chili Farmers

Henriyadi Henriyadi *, Vatcharaporn Esichaikul and Chutiporn Anutariya

ICT Department, School of Engineering and Technology, Asian Institute of Technology, Khlong Luang District, Pathum Thani 12120, Thailand; vatchara@ait.ac.th (V.E.); chutiporn@ait.ac.th (C.A.)
* Correspondence: st118501@ait.ac.th; Tel.: +66-63-376-1015

**Abstract:** Farm Management Information Systems (FMIS) assists farmers in managing their farms more effectively and efficiently. However, the use of FMIS to support crop cultivation is, at the present time, relatively expensive for smallholder farmers. Due to some handicaps, providing an FMIS that is suitable for small-holder farmers is a challenge. To analyze this gap, this study followed 3 steps, namely: (1) identified commodity and research area, (2) performed Farmers' Information Needs Assessment (FINA), and (3) developed the conceptual model using the Soft System Methodology. Indonesian smallholder chili farmers are used as a case study. The most required information of smallholder' farmers was identified through a qualitative questionnaire. Despite this, not all identified information needs could be accurately mapped. Thus, this indicates the need for a new FMIS conceptual model that is suitable for smallholder farmers. This study proposes an FMIS conceptual model for farm efficiency that incorporates five layers, namely farmers' information needs, data quality assessment, data extraction, SMM (split, match and merge), and presentation layer. SMM layer also provides a method to comprehensively tackle three main problems in data interoperability problems, namely schema heterogeneity, schema granularity, and mismatch entity naming.

**Keywords:** farm management information system; farmers' information needs assessment; soft system methodology; smallholder farmers; conceptual model; Indonesian chili farmers

## 1. Introduction

### 1.1. Background

The Farm Management Information System (FMIS) is a tool to assist farmers in managing their farms more effectively and efficiently. FMIS is a system that deals with the accuracy of data, optimization of the use of available resources, and processes by using advanced technologies for cultivating the farm [1]. Some researchers propose approaches to improving functionalities, such as improving management systems' functionality, interoperability, database inter-networking, and improving software architecture [2]. In the primary studies, 14 FMIS features appeared more than 7 times and 11 FMIS barriers appeared 3 times or more [3]. By accurately using FMIS, farmers can manage their farms more effectively and efficiently [4]. The main characteristics of existing FMISs are tailor-made applications that offer advanced functionality, focus on large farms, and concentrate solely on the specific needs of the users [3]. Moreover, over 75% of existing FMIS applications require a dedicated desktop computer to operate [5], rendering the current FMIS application expensive, especially for smallholder farmers. Therefore, providing an FMIS at an affordable price for smallholder farmers is challenging [6].

We should pay attention to smallholder farmers when developing an FMIS application due to some reasons. At present, there are about 570 million farms globally, of which, more than 475 million are smallholder farmers [7,8]. Smallholder farmers have common characteristics, including: (a) occupying a farm smaller than 2 ha [7], (b) the use of traditional

product market chains, (c) the use of family labor on the farm, (d) have on-farm activities as their only source of income, (e) employ the traditional farming system, (f) have limited financial support, and (g) operate without a form management system [9]. Thus, the application of FMIS by smallholder farmers is impossible considering their lack of knowledge as well as the costs and constraints associated with it. Providing an FMIS application that is suitable for smallholder farmers at an affordable price is a herculean task [10]. This study aimed to answer the main question: "how to develop an FMIS conceptual model that is applicable for smallholder farmers?" There are 3 main goals in this research, namely (1) identification of the smallholder farmers' information needs, (2) mapping information need into the existing FMIS conceptual model to find the gap between the existing conceptual model with the information needed, and (3) develop an FMIS conceptual model for smallholder farmers.

### 1.2. Previous Work

The Farmers' Information Needs Assessment (FINA) is a common method for identifying farmers' information needs. Various studies using FINA have already been conducted [11–19]. In this method, the collection of information required is grouped based on some criteria, making it more understandable [11,12,17]. However, the literature study conducted did not find any relevant research that uses the agribusiness subsystem in grouping information. In addition, no research was found to have explicitly used a qualitative approach in data collection, even though it generates various benefits over the quantitative approach [9]. For farmers' information need assessment, the qualitative approach promises some benefits such as obtaining an in-depth understanding of what information is most needed by farmers and finding indigenous information that was often unthinkable before.

Furthermore, many researchers have presented various perspectives concerning FMIS. Some authors highlighted its technical aspects [1,20] while others underscored the non-technical aspects [5,21,22]. These discussions, however, have proven to be restricted as they failed to consider smallholder farmers. To address this gap, this study proposes a conceptual model for Small Farm Management Information System (sFMIS) that is relatively distinctive compared to the existing FMIS conceptual model.

The development of the sFMIS conceptual model adheres to three principles. Firstly, it only provides the functionalities required by smallholder farmers to reduce the development cost. Secondly, it optimizes the use of open external data sources to reduce operational costs. Finally, it is available as a mobile-based application to reduce equipment expenditure.

However, deploying sFMIS can be challenging and presents many problems. One main drawback is understanding the information needed by farmers. Handling data interoperability problems relating to the use of external data sources is another challenge. Some problems associated with data interoperability are schema heterogeneity, schema granularity, entity naming mismatch, and data type mismatch [10]. Much research to date has tried to address the data interoperability problems associated with the usage of external data sources. There are three main problems in data interoperability at the schema level, namely schema heterogeneity, granularity data, and inconsistency field naming. Some researchers tried to tackle the problems by using the ontology matching approach. AgreementMaker [23], COMA++ [24], Cupid [25], Falcon-AO [26], and S-Match [27] are the most commonly used and discussed ontology matching approaches in the literature. Other researchers, on the other hand, tried to address the data interoperability problems using the database approach [28–30]. Despite these initiatives, no study so far has integrated all three obstacles that may arise in using external sources comprehensively.

Another thing that should be considered in application development is the adoption of new technology or application to target users. A new application is useless if it is not adopted by the target users. Indeed, introducing a new application to small farmers is not an easy challenge. Some aspects play an important role in adopting the new application, on the adopters' side, namely: technologies' technical features, users' perceptions (farmers and

farm employees) of innovation attributes, and users' characteristics such as age, education level, and existing computer skills [31].

Indonesia is an agricultural country. There are 27,682,117 agricultural households in Indonesia and approximately 10,104,682 of them work in the horticulture sub-sector [32]. However, from the literature study conducted, no research has been found related to the use of FMIS in supporting farming in Indonesia. Several earlier studies focused on certain aspects of the FMIS ecosystem. For example, some studies focus on monitoring plant growth [33], climate and planting calendars [34], water management [35], the use of drones [36], and related product marketing [37]. In addition, there is no mobile application (Android) that offers the use of FMIS as an agribusiness ecosystem. Existing applications are related to cultivation (SIPINDO, MyAgri, Lumbungin, Digitani), pest control (drtania, Plantix), marketing (tanihub, sayurbox), and financing (iGrow).

## 2. Methodology

The methodology used in this study is a combination of several methods, namely the purposive sampling method in selecting the commodity and research area, simple random sampling in selecting respondents, Farmers' Information Needs Assessment (FINA) in identifying information mostly needed by farmers, and Soft System Methodology (SSM) in developing the conceptual model. The block diagram of the methodology is presented in Figure 1, as explained as follows:

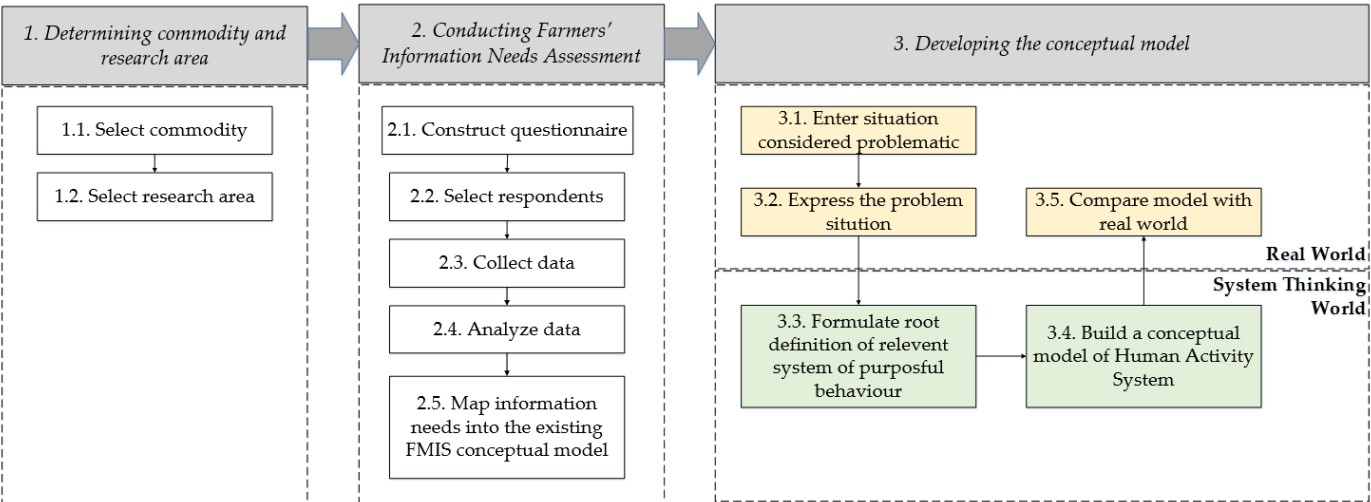

**Figure 1.** Research Methodology.

### 2.1. Determining Commodity and Research Area

The first step was determining the commodity and research area. Since the scope of the agricultural sector is extensive, various commodities or different agroecosystems require distinct handling methods. However, no method is suitable for all. Therefore, this study employed the case study approach. There are two activities in this step, namely the select commodity and the select research area.

#### 2.1.1. Select Commodity

The purposive sampling method was applied to select the commodity. The commodity selection is based on two important considerations: it is a seasonal crop with a life cycle of fewer than six months, and it is traditionally cultivated by the majority of smallholder farmers in Indonesia.

Chili (*Capsicum annum* L.) was used as a case study because it is one of the strategic commodities in Indonesia. The need for chili in Indonesia continues to grow with the increase in income and population [38]. Additionally, chili farming ensures high profits in a relatively short time. Chili plants are ready to be harvested from the age of 3–4 months

and the harvesting can be performed once a week until the plants are 6–7 months old [39]. Moreover, chili cultivation has a benefit to cost (B/C) ratio of 2.4 [39]. Despite the promising high profits, chili farming in Indonesia confronts a diverse range of challenges. Most Indonesian chili farmers are smallholder farmers, with each farmer cultivating only less than 0.5 hectares. They face a lack of information on new cultivation technology and in handling pests and diseases that often damage chili crops problems. They also struggle with high price fluctuations resulting from changes in supply and demand, and limitations in production capital and access to financing sources.

### 2.1.2. Select Research Area

The selection of research area started with selecting a province and was followed by selecting a district and sub-district. The purposive sampling method was employed. To select a province as a case study, statistical data analysis of the plantation area and average annual contribution to national chili production were considered. Thus, the West Java province was chosen based on the data obtained from the Central Bureau of Statistics [32,33]. The districts in this study, on the other hand, were selected based on three criteria: harvest area, distance to the research location, and willingness of the key informant. Thus, Sukabumi and Bandung Barat Districts were chosen as the research areas.

### 2.2. Conducting Farmers' Information Needs Assessment

The following step is assessing the farmers' information needed. This step includes four activities, namely: constructing a qualitative questionnaire, selecting respondents and conducting data collection, analyzing data, and mapping information needs into the existing FMIS conceptual model.

### 2.2.1. Construct a Qualitative Questionnaire

The first step in FINA was to construct a semi-structured qualitative questionnaire to elicit in-depth responses from the selected respondents in the study. The questionnaire was divided into two sections: the farmers' characteristics, and the farmers' information needs. The respondents' characteristics section consists of three groups of information: respondent's profile, ownership of the mobile phone, and a willingness to try a new application. Moreover, the farmers' information needs were categorized into five aspects following the agribusiness subsystems, namely: pre-planting, planting, harvesting, marketing, and supporting.

### 2.2.2. Select Respondents and Conduct Data Collection

The respondents were selected using simple random sampling. A total of 50 farmers were selected as the respondents, of which 27 farmers were from the Suntenjaya, a sub-district of Bandung Barat district, and 23 farmers from the Caringin and Kadudampit, a subdistrict of Sukabumi district.

The data collection was started with an explanation of the aim of the study and guidelines on how to fill out the questionnaire. The process continued with filling out the questionnaire by the respondents. In the end, an in-depth discussion with the key informant and some farmer representatives was carried out to collect more data.

### 2.2.3. Analyze the Data

The farmers' profiles were analyzed using the distribution frequency method. The analysis process started with grouping the data based on the "interval" or "bin" that had already been defined. The Data Analysis tools provided by Microsoft Excel were used to calculate the distribution frequency for each internal data.

The farmers' information needs were examined using word frequency analysis. Firstly, all collected data were divided into a list of words or terms using the text preprocessing method. In detail, text preprocessing consists of five consecutive processes: splitting into words, tokenizing, finding the root of the word, dropping unrelated words or terms, and

grouping similar terms. After that, the frequency for each word/term is conducted by computing the occurrence frequency for each word or term and sorting the occurrence frequency from highest to lowest score.

### 2.2.4. Map Information Needs into the Existing FMIS functionality

The activities in this step include determining FMIS conceptual model as a reference, mapping into the existing reference model, and finding the gap between information needs with the reference model. All activities were executed manually. This study uses the FMIS conceptual model by Sorensen [40] as a reference conceptual model. The output produced in this activity formed the basis for deciding whether to apply the existing conceptual model or to propose a new one.

### 2.3. *Developing the Conceptual Model*

The next step is developing the conceptual model using Soft System Methodology (SSM). SSM is a cyclical learning system that utilizes different human activities to investigate the actors in the real-world problem situation, how they perceive that situation, and their readiness [41]. The main aim of this step is to decide on the appropriate activity, taking into account the perceptions, judgments, and values of various actors [42]. Although SSM originally consists of seven stages [41], it is not necessary to follow all of the phases [43]. For this study, only the first five stages of SSM were adopted to develop the model.

### 2.3.1. Identify the Existing Problem Situation

This first stage in SSM is to find out the problem situation and understand what the system is. Any possible problems that may be encountered were identified. This process was conducted through desk study and group discussion. The output of this step is all problems that may arise with the system that need to develop.

### 2.3.2. Convert the Problem Situation into a Structured Problem

The following stage is converting the problem situation into becoming structured problem. This was executed by organizing the unstructured problems that were already identified into structured problems. The mnemonic CATWOE (Customer, Actors, Transformation Process, Worldview, Owners, Environmental constraint) analysis method was employed to organize. The result of the CATWOE analysis was translated into a "Rich Picture", presenting a whole picture of the developed system.

### 2.3.3. Formulate Root Definition of Relevant System

The structured problems were analyzed to find the system under investigation using a root definition approach. A root definition is a sentence that describes the ideal system: What does the system do? How does it function? What is its purpose?

### 2.3.4. Build a Conceptual Model of the Human Activity System

The main purpose of this stage is to produce a model of what the system should execute. Data gathered from earlier works and previous models are used as references in the development of the conceptual model. The outcome is an FMIS conceptual model for smallholder farmers with Indonesian chili farmers as a case study.

### 2.3.5. Compare the Conceptual Model with the Identified Problem Situation

The final stage ensures that the identified problems have been addressed in the conceptual model, using a diligent mapping process for each part of the conceptual model into each identified problem.

## 3. Results

This study uses chili commodities as a case study. The 50 smallholder farmers from Bandung Barat and Sukabumi District, West Java province, were selected as respondents. Quantitative analysis of the respondents' characteristics indicates that the target respondents from the study are following the objectives of the study. Moreover, through the word frequency analysis of the questionnaire, the most information needed by smallholder farmers was identified. Additionally, through qualitative methods and in-depth discussion, some information that was not thought of before was found. However, not all identified information needs could be precisely mapped into the existing FMIS conceptual model. Therefore, a new FMIS conceptual model for smallholder farmers was proposed. Different from the existing conceptual model, the proposed model focuses on utilizing as much as possible external data sources, can handle data interoperability problems that may occur, and provides Android as an application interface platform.

### 3.1. Farmers' Information Needs

3.1.1. Analyzing the Respondent's Characteristics

Three groups of information in respondents' characteristics were included in the analysis, namely: respondents' profile, mobile phone ownership, and a willingness to try a new application. The descriptive analysis was used to analyze the respondents' characteristics. The analysis showed that the majority of respondents are smallholder farmers, they have a mobile phone and are willing to install a new application under some conditions.

Respondents' Profiles

The basic demographic features of respondents are shown in Table 1. The respondents had a mean age of about 38 years old and were mostly composed of low-level educated individuals, smallholder farmers, and renters of farmland for their cultivation. Approximately 78 percent of the respondents cultivated rented farmland, and 20% cultivated on their owned farmland. Additionally, the respondents cultivate on land with an average area of roughly 0.6 hectares, in a range between 0.1 and 2 hectares. However, the respondents had an average of 14 years of on-farm experience, in the range of 1 to 35 years.

**Table 1.** Demographic characteristics of respondents.

| Variable | Total Respondents $n = 50$ | $p$-Value |
|---|---|---|
| Age (mean ± SD, range (in years old)) | 37.6 ± 9.7 (16–57) | <0.01 |
| Education Level ($n$, %) Primary school Secondary School High school Bachelor | 25 (50%) 16 (32%) 8 (16%) 1 (2%) | <0.01 |
| Experience (mean ± SD, range, in years) | 13.5 ± 9.2 (1–35) | <0.01 |
| Land ownership ($n$, %) Owned Rental Owned and rental | 10 (20%) 39 (78%) 1 (2%) | <0.01 |
| Cultivation area (mean ± SD, range, in Ha) | 0.57 ± 0.48 (0.1–2) | <0.01 |

Mobile Phones Ownership

The mobile phone ownership of the respondents is shown in Table 2. It was found that 72% of the respondents owned a mobile phone, mostly an Android phone. In addition, approximately 48 percent of respondents (or 77 percent of those using an Android phone) subscribed to a monthly internet subscription, with slightly more than half of them (58%) spending over Rp. 50,000 (US $4).

**Table 2.** Mobile phone ownership of smallholder chili farmers in Sukabumi and Bandung Barat districts.

| Variable | Criteria/Range | Frequency | Percentage |
|---|---|---|---|
| Mobile phone ownership | No | 14 | 28% |
| | Yes | 36 | 72% |
| | Total | 50 | 100% |
| Mobile phone operating system | Android | 31 | 62% |
| | Feature phone | 5 | 10% |
| | No phone | 14 | 28% |
| | IOS | 0 | 0% |
| | Total | 50 | 100% |
| Subscription to an Internet package | No | 7 | 14% |
| | Yes | 24 | 48% |
| | No phone | 14 | 28% |
| | Feature phone | 5 | 10% |
| | Total | 50 | 100% |
| Average expenditure on monthly data package (in IDR) | 0–25,000 | 3 | 6% |
| | 25,001–50,000 | 7 | 14% |
| | 50,001–75,000 | 6 | 12% |
| | 75,001–100,000 | 8 | 16% |
| | No phone | 14 | 28% |
| | Not support | 5 | 10% |
| | Total | 50 | 100% |

Willingness to Try a New Application.

Another important aspect of the respondent's profile is their willingness to try a new application. It was found that 97.84 percent of the respondents considered trying new applications if they met certain criteria, such as the application supporting their agricultural farming activities, the application providing direct discussion to experts/extension workers, and the application providing facilities for marketing their product. Analysis of data also revealed three main factors influencing farmers to try new applications: ease of installation and use; benefits they obtain; new experiences in using technology.

3.1.2. Analyzing the Farmers' Information Needs

The specific information needs of the farmers were assessed based on their responses using a word frequency method. There were 1298 pieces of information derived from splitting, tokenization, and finding the root of the word. The details of all identified information were grouped based on the similarity terms into 32 types of information needed are presented in Appendix A Table A1.

Furthermore, all of the required information was sorted based on the occurrence frequency to find the top ten information mostly needed by smallholder farmers. The in-

depth discussion led to additional required information. All identified farmers' information needed is presented in Table 3.

**Table 3.** The smallholder farmers' information needs in Sukabumi and Bandung Barat Districts.

| Source of Data | Information Needed | Occurrence Frequency |
|---|---|---|
| Word frequency analysis | cultivation technology | 233 |
| | market price | 128 |
| | agricultural financing | 106 |
| | land preparation | 97 |
| | consultation | 96 |
| | market demand | 68 |
| | handling pest and disease | 65 |
| | another region with the same crop | 60 |
| | seed description | 59 |
| | weather forecast | 54 |
| In-depth discussion | farmland location | n.a |
| | Farmland owner | n.a |
| | the existing crop that is being cultivated | n.a |
| | financial record keeping | n.a |
| | recording their cultivation activities | n.a |

The "n.a" in occurrence frequency indicates that the information needs are obtained from the in-depth discussion with representative farmers process and do not obtained from the qualitative questionnaire that the respondents filled in. The in-depth discussion was conducted with the "key informant" or "pioneer' farmer" and several senior farmer representatives, namely farmers who have more than 10 years of farming experience. With in-depth discussion find information or idea that was not thought of before.

3.1.3. Mapping Farmers' Information Needs into Existing FMIS Functionalities

Matching and mapping each identified information need with the reference conceptual model was performed manually. The result of the mapping process presented in Figure 2 indicated that some information needs could not be mapped exactly on the referenced FMIS conceptual model [40]. This demonstrates that not all functionalities provided in the FMIS conceptual model required smallholder farmers; thus, establishing the importance of a new conceptual model for the farmers to fully benefit from FMIS.

*3.2. Develop a Conceptual Model*

The result of the farmers' information needs assessment was used to develop the conceptual model. The proposed conceptual model consists of five layers, namely: farmers' information needs layer, assess the data sources quality layer, data extraction layer, split-match-merge layer, and presentation/user interface layer. The detailed processes of developing the conceptual model are as follows.

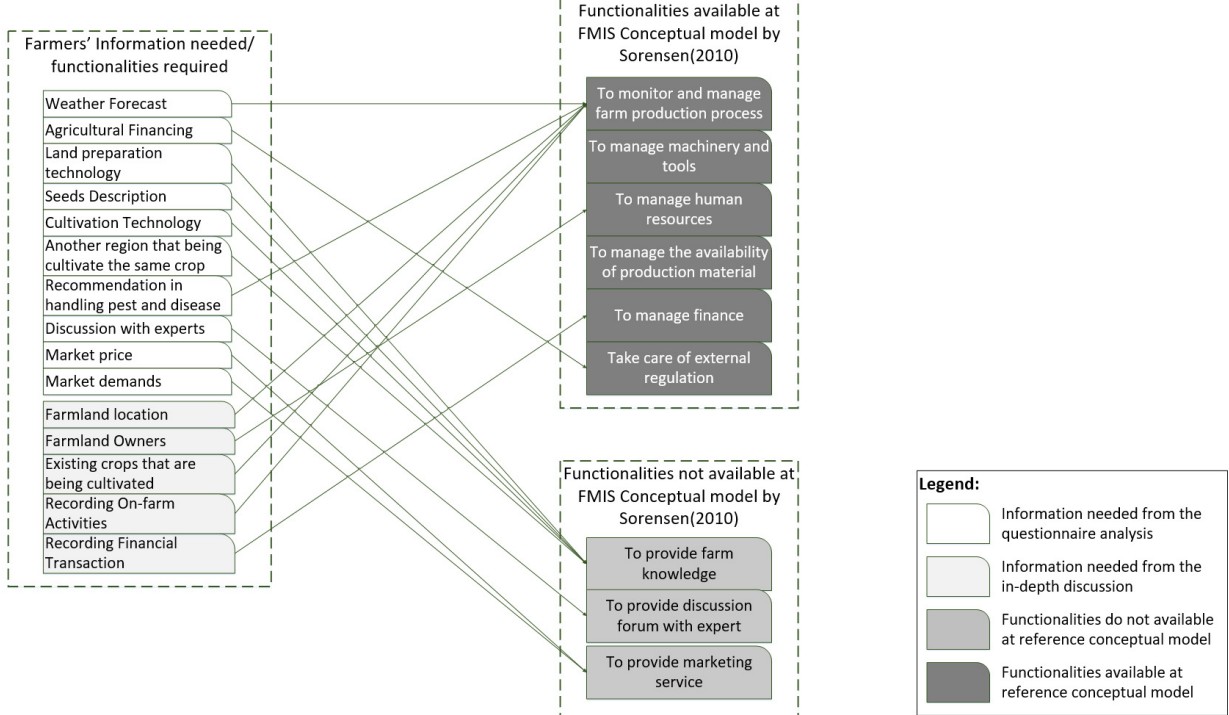

**Figure 2.** Mapping smallholder farmers' information needs into the FMIS Conceptual Model.

### 3.2.1. Identifying the Existing Problem Situation

As explained in the background section, providing an FMIS to smallholder farmers has various challenges, including requiring a comprehensive understanding of the information preferred by farmers and handling data interoperability problems associated with the use of external data sources. Based on these challenges, several critical questions are considered in providing sFMIS:

- Where do the data sources for each piece of information come from? Is this data available online?;
- What is the quality of each candidate's data source? Are all external data sources eligible for extraction, transformation, and loading?;
- Data sources are available in many formats, what method is used to extract each data source into a temporary database?;
- What is the process of transforming and loading data from the temporary database into the application database? How the algorithm could tackle the data interoperability problems that may arise in transforming and loading data process?;
- How can information be presented to users in an easy, inexpensive, and user-friendly way?

### 3.2.2. Converting the Problem Situation into a Structured Problem

The unstructured problems identified earlier are organized. When using external data sources, the following functionalities should be provided:

- External data source quality assessment;
- Data extraction for each eligible candidate's external data source into the temporary database;
- Data loading and transformation can handle data interoperability problems that may arise.
- A friendly User Interface (UI).

Furthermore, the CATWOE method was employed, and the following items were obtained:

- Customer: the primary actor of this model is the farmer, and the secondary actors are traders, experts/extension workers, and local government officers;
- Actors: the primary actors of this system are external websites that supplied data to support the android application. Whereas secondary actors are a group of users that interact with the android application, such as farmland owners, farmers, traders, and other data providers;
- Transformation process: in collecting and inputting data, manually inputting data transformed into an automatic process through extracting and loading data from many external data sources;
- Worldview: external data sources that can potentially be reused by the system to help farmers decide on aspects related to their farm;
- Owners: the primary owner of this system is the researcher who develops the system, while the secondary owners are the organizations who implement and manage the system;
- Environmental constraints: the primary constraints in developing the systems are the quality of data provided by an external website and access rights to external data sources. Whereas the secondary constraints are quality of infrastructure. Minor constraints are the quality of network or Internet infrastructure when collecting data from external data sources.

Moreover, to have a whole view of the system, the result of CATWOE analysis was drawn into a "Rich Picture" as shown in Figure 3. The core element of this rich picture is an sFMIS android application with four main customers/targeted users, namely farmers, farmland owners, traders, and experts/extension workers. The android application has support data from the application database through an API (Application Programming Interface) service platform. There are two data sources for the application database, namely manual data entry through the application e-form and data as a result of the extract, transform, and load (ETL) process from the temporary database. The ETL process also handles the data interoperability problem that may occur; whereas the temporary database itself is a container of the data extraction process from many external data sources.

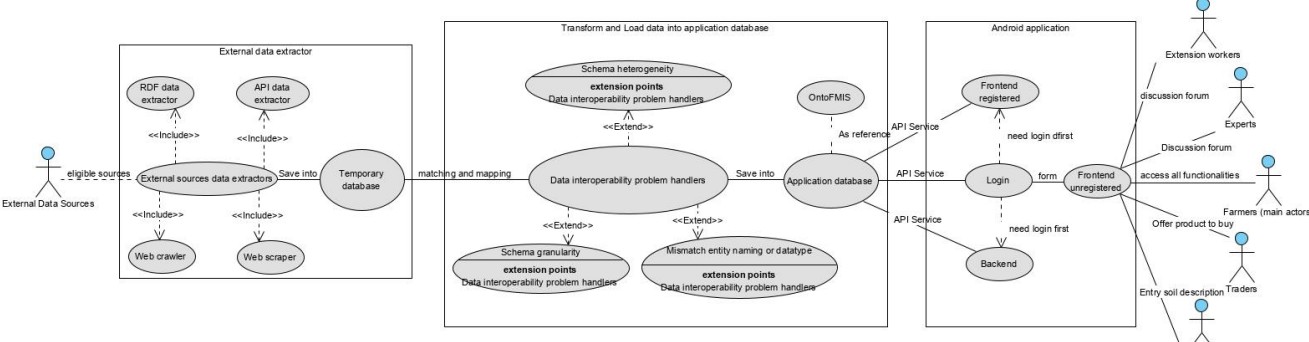

**Figure 3.** A rich picture in UML diagram format of the small Farm Management Information System (sFMIS).

### 3.2.3. Formulating the Root Definitions of Relevant Systems

The next stage is formulating a root definition for the relevant system performed by answering the three main questions as follows:

- What does the system do? The system can collect data from many external websites and conduct the assessment of the quality of data sources, extract data, transform them and load them into the application storage using the Android application;
- How does it function? The system will perform some functions, starting with identifying candidate sources of data, collecting data from many external websites, assessing the quality of the data sources, extracting, transforming, and loading data into the

application storage, and providing an android application as the interface between system and users;

- What is its purpose? The system should be easy to use by smallholder farmers. Moreover, this system should provide information deemed necessary by users. Additionally, the system should have the capability to interact with smallholder farmers, especially in managing their crop production process;

3.2.4. Building a Conceptual Model of the Human Activity System

After understanding the root definitions for the system, the next stage is creating a conceptual model. All of the data collected in the earlier steps were compiled and analyzed to develop the conceptual model. The result of developing the sFMIS conceptual model for Indonesian Chili Farmers as presented in Figure 4 consists of five layers, namely: (a) farmers' information needs layer, (b) assess the data quality layer, (c) data extraction layer, (d) split, match and merge layer and € presentation/user interface layer. The detailed explanation for each layer is as follows:

a　　Farmers' Information Needs Layer

This layer consists of the list of farmers' information needs is the result from the FINA as shown in Table 3. All of the information needs are a combination of the result of two analysis methods, word frequency analysis and an in-depth discussion summary with the key informant and two senior farmers.

b　　The data quality assessment layer

The second layer of the model involves assessing the data quality of all candidate data sources. The first activity in this layer is to identify the candidate data sources for each functionality. All identified candidate data sources for each sFMIS functionality are shown in Appendix B Table A2.

The candidate data sources for each functionality were retrieved by "googling" related keywords and other sources of information. Among 15 functionalities required, 9 functionalities found candidate external data sources and 6 functionalities required manual data entry. Furthermore, the process continued with assessing the quality of data sources. This is a fundamental process because data from various external sources have a variety of formats, platforms, levels of detail, and ownership models. Several researchers have proposed assessment dimensions to evaluate data quality [44–49]. For this study, eight of the most significant assessment dimensions from a combination of several references are presented in Table 4. The weighting score for each dimension was calculated using AHP (Analytical Hierarchy Process) method with a consistency index (CI) = 0.060395782 and consistency ratio = 0.042833888. The detailed scoring criteria for each dimension are also shown.

**Table 4.** The eight dimensions of data quality assessment in Small Farm Management Information System.

| No. | Dimensions | Weighting Score | Scoring Criteria | | | | |
|-----|------------|-----------------|------|------|------|------|------|
| | | | 5 | 6 | 7 | 8 | 9 |
| 1 | Accessibility | 0.28 | protected | login and sent via email | login and download file | free with key | free access |
| 2 | License | 0.22 | copyright | limited free for registered user | free for registered user | free limited service | free |
| 3 | Source reliability | 0.17 | personal blog/others | others' company | Other organization | well-known company | Government/ international org |
| 4 | Connectedness | 0.13 | others | pdf | html | XLS/csv | API/RDF |
| 5 | Accuracy | 0.09 | very low | low | medium | high | very high |

**Table 4.** *Cont.*

| No. | Dimensions | Weighting Score | Scoring Criteria | | | | |
|---|---|---|---|---|---|---|---|
| | | | 5 | 6 | 7 | 8 | 9 |
| 6 | Completeness | 0.06 | 20% | 40% | 60% | 80% | 100% |
| 7 | Format Consistency | 0.04 | not use standard, inconsistent | not use standard, inconsistent | not use standard, consistent | use standard, inconsistent | use standard, consistent |
| 8 | Timeliness | 0.02 | never | seldom | sometime | often | always |

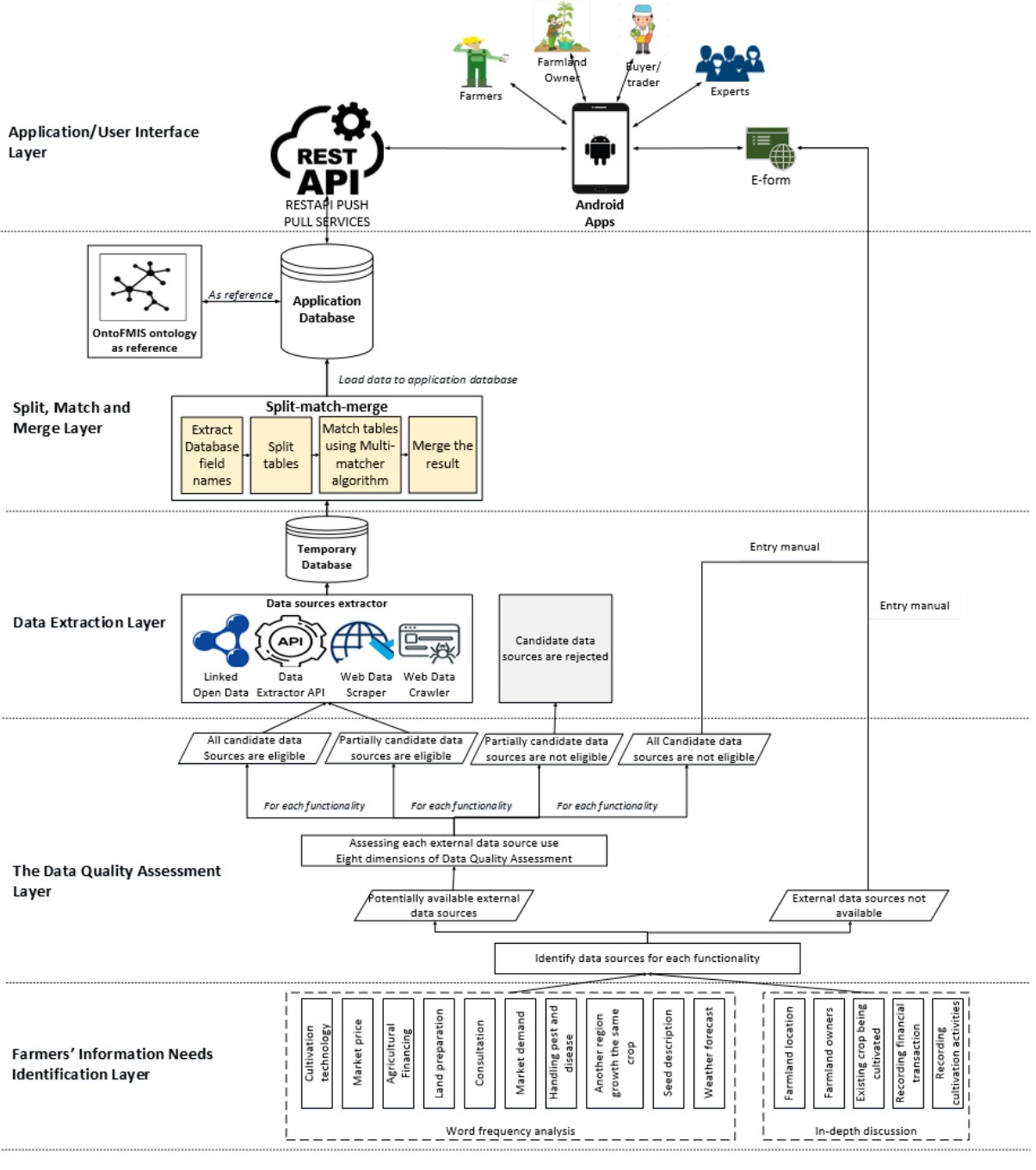

**Figure 4.** The small Farm Management Information System Conceptual Model for Indonesian Chili Farmers.

Moreover, assessing the data quality for each candidate's external data source uses a multi-criteria decision-making (MCDM). Appendix C Table A3 presented the details assessment score for each candidate's external data source. Among 35 candidate external data sources, 26 were accepted and eligible as external data sources, and 9 were rejected. However, some accepted data sources require manual intervention before they could be used, such as seed description and handling pest disease. Additionally, three types of candidate external data sources were rejected: those with un-supported data format, granularity data, and access-rights problems. Most of the data coming from the android application were rejected due to technical constrains.

c    Data extraction layer

The data extraction layer consists of four data extraction methods depending on the format of the data provided. The first one is Linked Open Data (LOD), a tool used to data extract the external data sources that are available in RDF format. The second is the data extractor API, a tool used to extract data from the external data sources that provide the API service. The third method is a web crawler, which draws out the external data source that is available on static pages. The last one, the web scraper, extracts the external data source that is available on dynamic pages. The output of data extraction processes will then be saved in temporary storage for the next layer to transform and load the data. The flowchart to select the data extraction method based on the data provided by external data sources is presented in Figure 5.

d    Split, match, and merge layer

The following layer in the conceptual model involves splitting, matching, and loading the data. This layer is the most important in this conceptual model since the quality of the information provided to the users depends on it. Data are loaded from the temporary database into the application database, as illustrated in Figure 6.

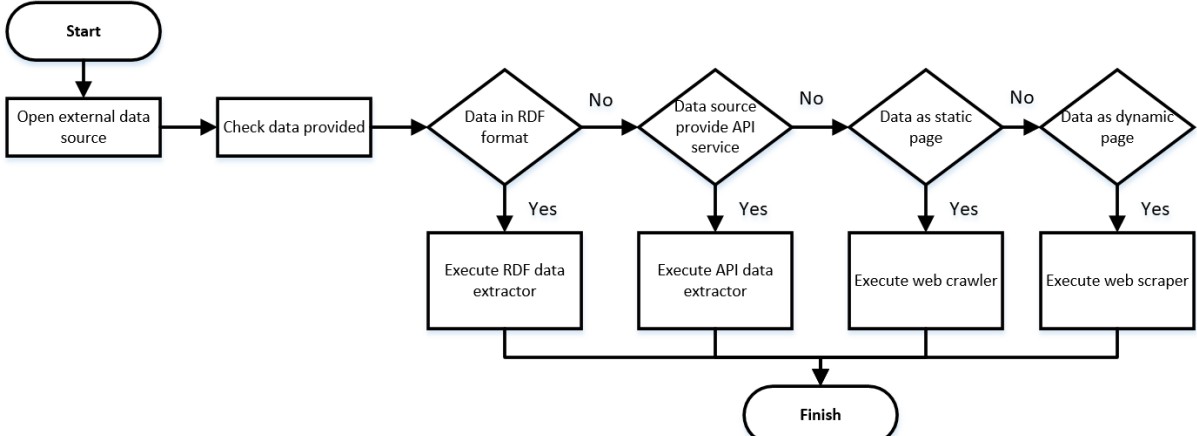

**Figure 5.** The flowchart in selecting data extractor tools.

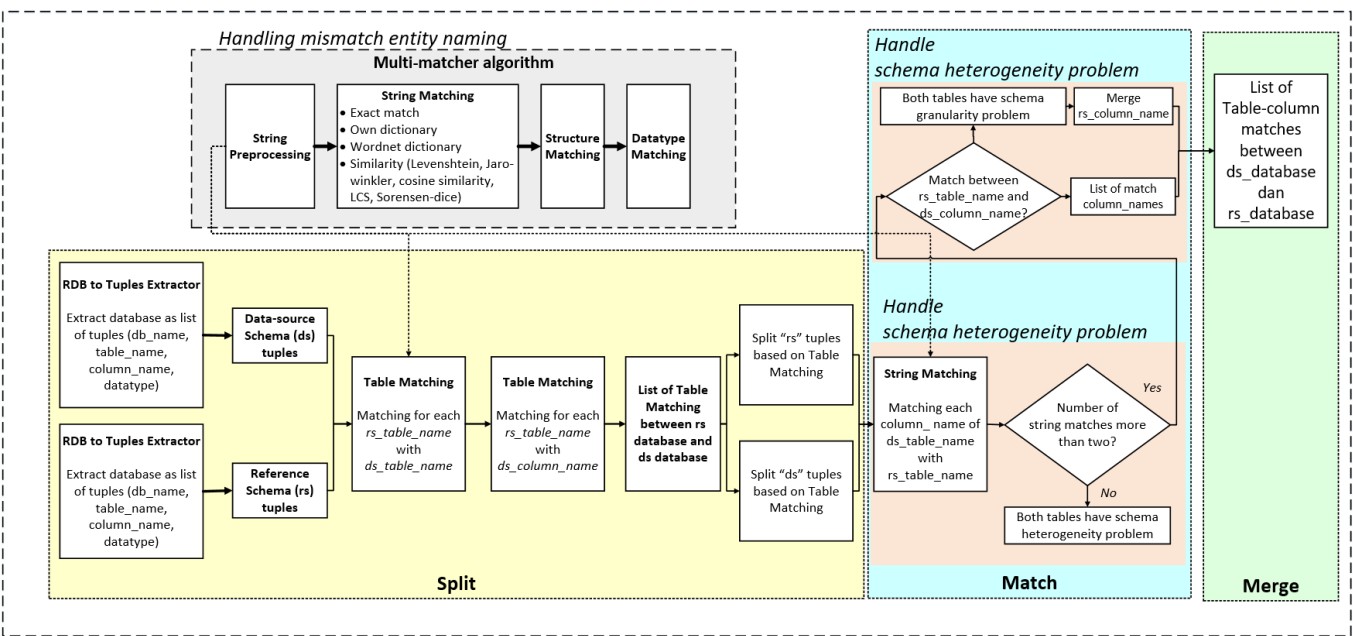

**Figure 6.** The details of the split, match and merge method with a multi-matcher algorithm.

The "split-match-merge method with multi-matcher algorithm" [10] was employed for this step. This layer consists of a series of activities. The first activity is converting both application and data source databases into a set of tuples. The application database developed referred to the OntoFMIS (http://103.169.28.91/ontofmis/, accessed on 22 March 2022), an ontology for small farm management information systems. Furthermore, the following activity is class matching between data source table_name and reference table_name. The next activity is is extracting each schema based on the result of class matching. For each class matching, the process continued with column name matching and datatype matching. Moreover, this layer also handles data interoperability problems that may arise, such as schema heterogeneity, the granularity of data, mismatch data type, and mismatch field naming. The output of this layer is an application database that is ready for Android applications to supply data to users.

The split-match-merge method is a modification of the method proposed by Wickham [50]. This method is aimed at making the matching and mapping process run effectively and efficiently. Through this method, the matching and mapping process only executes one-to-one matching between temporary database table structure application database table structure. While the multi-matcher algorithm is the heart of this model. The quality of data presented to the users depends on the proses inside the multi-matcher algorithm. A combination of hybrid and composite from many matcher algorithms was used to construct this algorithm. All data interoperability problems that may arise are tackled using this algorithm. Matcher algorithms that executed cascade in this paper are as follows:

- Exact-matcher: a matcher to find whether two words exactly match;
- Word synonym: a matcher to check whether two strings are synonyms. A total of 3 sources were used as of dictionary, namely (1) Wordnet in English, (2) Wordnet in the Indonesian Language, and (3) creating own dictionary;
- Similarity matcher: a matcher to find the similarity of two terms. This study uses a hybrid model as a combination of five string-similarity algorithms, namely: (1) Levenshtein, (2) Jaro-Winkler, (3) Cosine similarity, (4) Longest Common Substring (LCS), and (5) Sorensen-dice. The result for each algorithm was normalized and following with calculation the average similarity with the acceptance score is 0.9.

Moreover, the evaluation performance of the 'split-match-merge method with multi-matcher algorithm" to tackle the three data interoperability uses the precision, recall, and accuracy indicators. The result of evaluating the algorithm to handle each data interoperability problem is as follows:

- Mismatch entity naming. The mismatch entity naming is conducted by matching each entity naming of the temporary database with all entity naming of the application database. There are 236 terms extracted from the temporary database and 251 terms were extracted from the application database. Table 5 present the calculation confusion matrix for handling the mismatch entity naming problem.

**Table 5.** Calculation TP, TN, FP, and FN in handling mismatch entity naming.

| | | Manual Checking | |
| | | True | False |
|---|---|---|---|
| A split-match-merge method | Positive | TP = 138 | FP = 18 |
| with a multi-matcher algorithm | Negative | TN = 59,068 | FN = 4 |

The accuracy, precision, and recall calculation consecutively as follows

$$Accuracy = (TP + TN)/(TP + FP + FN + TN)$$

$$= 0.88$$

$$Precision = TP/(TP + FP)$$

$$= 0.97$$

$$Recall = TP/(TP + FN)$$

$$= 0.94$$

- Schema heterogeneity problem. The handling schema heterogeneity problem is executed through two steps, matching the table name, and matching the entity naming for each matched table. Table 6 presents the calculation of the confusion matrix in handling the schema heterogeneity problem.

**Table 6.** Calculation TP, TN, FP, and FN in handling schema heterogeneity problem.

| | | Manual Checking | |
| | | True | False |
|---|---|---|---|
| A split-match-merge method | Positive | TP = 186 | FP = 6 |
| with a multi-matcher algorithm | Negative | TN = 6874 | FN = 8 |

The accuracy, precision, and recall calculation in a row are as follows

$$Accuracy = (TP + TN)/(TP + FP + FN + TN)$$

$$= 0.98$$

$$Precision = TP/(TP + FP)$$

$$= 0.97$$

$$Recall = TP/(TP + FN)$$

$$= 0.96$$

- Schema granularity problem. The handling schema granularity problem is executed by matching the entity naming of the table of the application database into the table name of the temporary database and matching the entity naming of the table of the

temporary database into the table name of the application database. The confusion matrix in handling schema heterogeneity problem is presented in Table 7.

**Table 7.** Calculation TP, TN, FP, FN in handling schema granularity problem.

| | | Manual Checking | |
| | | True | False |
| --- | --- | --- | --- |
| A split-match-merge method | Positive | TP = 88 | FP = 5 |
| with a multi-matcher algorithm | Negative | TN = 30,139 | FN = 12 |

The accuracy, precision, and recall calculation successively as follows

$$Accuracy = (TP + TN)/(TP + FP + FN + TN)$$

$$= 0.99$$

$$Precision = TP/(TP + FP)$$

$$= 0.946$$

$$Recall = TP/(TP + FN)$$

$$= 0.88$$

- Handling data interoperability problems comprehensively. Besides the performance in handling data interoperability problems individually, the algorithm performance evaluation is also conducted in handling data interoperability problems comprehensively. Table 6 present the result of the confusion matrix of TP, TN, FP, and FN.

The accuracy, precision and recall calculation consecutively as follows

$$Accuracy = (TP + TN)/(TP + FP + FN + TN)$$

$$= 0.998$$

$$Precision = TP/(TP + FP)$$

$$= 0.969$$

$$Recall = TP/(TP + FN)$$

$$= 0.959$$

e     Presentation/user interface layer

The top layer of this model is the presentation layer which consists of two main components. The first component is the API RESTful service server. This service is used the Android applications utilizing information provided by the application database. This component provides a push and pull service for storing and providing data. The second component is an android application which acts as an interface between the system and the users. Besides providing information, it also provides an e-form to enter data into the system. There are five target users for this Android application, namely: smallholder farmers as the main users, and four groups of the user as a supporting system, including experts, extension workers, traders, and soil specialists. Expert and extension workers have a role to answer all questions or requests for information raised by farmers. The trader's role is to submit information into the application if they need to buy chili stock owned by farmers or offer to buy chili from farmers, whereas a soil specialist is a person who appointed by authorized institutions to provide information related to the characteristics of farmland managed by farmers.

### 3.2.5. Comparing the Conceptual Model with the Identified Problem Situation

This stage aims to ensure that the conceptual model fully considers all of the problem situations that were identified. The first layer of the conceptual model relates to a list of farmers' information needs. Moreover, the second layer deals with the functionality of assessing the quality of the external data source of each candidate whereas the third layer handles the extraction of data from each eligible candidate's external data source into the temporary database. The fourth layer transforms and loads data into the application database. The last layer of the conceptual model deals with the functionality of a front-end application. Based on these facts, we can conclude that the model considers all of the identified problems.

## 4. Discussion

FMIS helps farmers to manage their farms effectively and efficiently. However, the existing FMIS application is relatively expensive for smallholder farmers [6,51]. This study proposed a new conceptual model of FMIS that is to satisfy the smallholder farmers' information needs [18]. The proposed conceptual model consists of five consecutive layers. To make an FMIS that is suitable for smallholder farmers, the development of the sFMIS conceptual model adheres to three principles. Firstly, it only provides the functionalities required by smallholder farmers to reduce the development cost. Secondly, it optimizes the use of open external data sources to reduce operational costs. Finally, it is available as a mobile-based application to reduce equipment expenditure.

What distinguishes it from the existing conceptual model is identifying farmers' information needs using qualitative approaches as the first layer of the model. This layer is important because the application aimed to provide the information needed by smallholder farmers. The use of a semi-structured qualitative questionnaire in this study revealed information needs that were not considered during the research process such as regions that grew the same crop and long-term weather forecast. The mapping of the farmers' information needs showed that not all information could be represented accurately with the functionalities of the FMIS conceptual model. Several modifications to the existing FMIS conceptual model are required to meet the needs of smallholder farmers. This also proof that a new FMIS conceptual model that applies to smallholder farmers is essential.

The second layer promotes the different methods on how to assess the quality of external data sources [45,47]. The eight dimensions that are equipped with assessment criteria for each dimension will make it easier for users to apply the developed conceptual model. Moreover, the third layer focus on the data extraction process from all eligible external data sources. Data extraction from many external data sources conducted in many ways depending on the data provided by external data sources. The REST API is used as the standard method to extract data. The web-crawling and web-scraping method is used to extract data if the external data sources do not provide API service.

The split, match and merge method are the most important layers of the conceptual model. The quality of data provided to end-users depends on the quality of data transforming in this step. This study uses a "split-match-merge method with multi-matcher algorithm" [10] to comprehensively address the three main data interoperability problems. Using a different similarity dictionary that is constantly evolving is the key factor that distinguishes the algorithm from others. Moreover, the algorithm employs a multi-matchers algorithm for term matching to improve the matching result. The algorithm that can tackle three main data interoperability problems is a differentiator from the current FMIS conceptual model. The evaluation of the algorithm performance using accuracy, precision, and recall indicators have shown that the algorithm could tackle the data interoperability problems very well. The calculation of the confusion matrix indicates that True Negative (TN) has a score which is too high compared with other indicators. This is happening because each term extracted from the temporary database only has potential matching with only one term extracted from the application database. On the other hand, the total

iteration is equal to all terms from the temporary database multiplied by all terms extracted from the application database.

The top layer is an Android application as an interface between the application and users. Choosing an Android application rather than a desktop-based application because the mobile performed better on user adoption, engagement, and retention [52]. Moreover, the usage of mobile phones will reduce the need for buying a desktop computer for running the application [5].

The proposed small farm management information system conceptual model was developed, explicitly based on the case study of chili farmers in West Java taking into account distinct requirements of external data sources for different commodities or different regions. Therefore, when applying the model to other commodities, some layers should be modified. Firstly, the information required for farmers' information needs assessment layer must be reestablished. Additionally, assessing the candidate external data source layer need to be reclassified. Another modification is required for the keywords in the dictionary used for improved results in transforming and loading data into the application database layer. Lastly, the functionality for SIMUSTI must be changed and adapted to the application layer.

## 5. Conclusions

The results from conducting the farmers' information need assessment produced a list of information most required by smallholder farmers. However, not all of the required information could specifically be mapped into the reference conceptual model of FMIS. This means that the smallholder farmers studied do not require some of the functionalities provided in the FMIS conceptual model. On the other hand, some functionalities required by smallholder farmers are not facilitated by the existence of the FMIS conceptual model proving that the requirement to develop an FMIS differs for smallholder farmers. This study, therefore, proposes a new FMIS conceptual model for smallholder farmers termed small Farm Management Information System (sFMIS). The provision of functionalities that meet the needs of smallholder farmers and the use of external data based on the data interoperability model are the most distinguishing features of sFMIS compared to the existing Farm Management Information System (FMIS). Therefore, the sFMIS ensures better farm management for smallholder farmers at affordable prices in application development and less operational costs. The sFMIS model uses Indonesian chili farmers as a case study and consists of five layers that can be applied to other commodities, with some modifications.

**Author Contributions:** Conceptualization, H.H., V.E. and C.A.; data collection, H.H.; methodology, H.H.; supervision, V.E. and C.A.; writing—original draft, H.H.; editing and correction, V.E. and C.A. All authors have read and agreed to the published version of the manuscript.

**Funding:** This research was fully supported by the Sustainable Management of Agricultural Research and Dissemination (SMARTD) Project (World Bank P117243)—Indonesian Agency for Agricultural Research and Development—Ministry of Agriculture.

**Institutional Review Board Statement:** Not applicable.

**Informed Consent Statement:** Not applicable.

**Data Availability Statement:** OntoFMIS, an ontology for small farm management information systems is temporarily available at: http://103.169.28.91/ontofmis/, accessed on 22 March 2022. Other data presented in this study are available on-demand from the first author at (st118501@ait.ac.th).

**Acknowledgments:** We thank our colleagues from the Indonesian Agency for Agricultural Research and Development (IAARD) and the Asian Institute of Technology (AIT), who provided insight and expertise that greatly assisted the research and prototype development.

**Conflicts of Interest:** The authors declare no conflict of interest. Funders have no role during the research process of this manuscript. The research process starts with the research design, collection, analysis, interpretation of data, writing a script, and publishing of the results.

## Appendix A

**Table A1.** The Frequency of Occurrence of Any Information Needed by Respondents in Sukabumi and Bandung Barat Districts.

| No. | Agribusiness Sub-System | Information Needed | Occurrence Frequency |
|-----|-------------------------|---------------------|----------------------|
| 1 | Pre-planting | agricultural financing | 106 |
|  |  | land preparation | 97 |
|  |  | another region with the same crop | 60 |
|  |  | seed description | 59 |
|  |  | machinery description | 46 |
|  |  | agricultural machinery | 23 |
|  |  | agricultural insurance | 15 |
|  |  | seed production technology | 9 |
|  |  | soil characteristic | 6 |
|  |  | seed availability | 5 |
|  |  | fertilizer and seed subsidies | 3 |
| 2 | Planting | cultivation technology | 233 |
|  |  | handling pest and disease | 65 |
|  |  | weather forecast | 54 |
|  |  | seed recommendation | 4 |
|  |  | labor availability | 2 |
| 3 | Harvesting | packaging | 22 |
|  |  | storage technology | 19 |
|  |  | grading | 5 |
|  |  | yield processing | 3 |
|  |  | Warehouse | 2 |
| 4 | Marketing | market price | 128 |
|  |  | market demand | 68 |
|  |  | marketing | 42 |
|  |  | transportation | 41 |
| 5 | Support | consultation | 96 |
|  |  | training | 40 |
|  |  | assistance | 20 |
|  |  | regulation | 11 |
|  |  | management | 10 |
|  |  | technical support | 4 |
|  | Total |  | 1298 |

## Appendix B

**Table A2.** Candidate Data Sources for Each Small Farm Management Information System (sFMIS) Functionality.

| No. | Functionalities | Candidate Data Sources (If Available) |
|---|---|---|
| 1 | Farmland location | Google Map, Open Street Map |
| 2 | Weather forecast | Open Weather Map, BMKG, World Bank |
| 3 | On-farm activities | Manual data entry |
| 4 | Handling pest and disease | Opete, IAARD, ICHORT, MyAgri, SIPINDO |
| 5 | Existing crops that are being cultivated | Manual data entry |
| 6 | Farmland owners' profile | Manual data entry |
| 7 | Financial transaction recording | Manual data entry |
| 8 | Agricultural financing | KUR (Kredit Usaha Rakyat)/people's business credit from Ministry Coordinator of Finance, Google News |
| 9 | Land preparation technology | Cyber extension, IAARD, Youtube |
| 10 | Seeds description | PPVT, IAARD, MyAgri, DBVaritas |
| 11 | Cultivation technology | Cyber extension, IAARD, Youtube, Repositori Publikasi, SIPINDO, ITani, Digitani |
| 12 | Another region cultivating the same crop | Generate by application |
| 13 | Consultation | Manual data entry |
| 14 | Market demand | Manual data entry |
| 15 | Market price | Toko tani BKP, PIHPS, Shopee, Bukalapak, Sayurbox, Tanihub, Info pangan Jakarta |

## Appendix C

**Table A3.** Data Quality Assessment Based on Eight Dimensions Criteria.

| No. | Functionality | Candidate External Data Source | Access w = 0.28 | Lice w = 0.22 | Sour w = 0.17 | Conn w = 0.13 | Accu w = 0.09 | Comp w = 0.06 | Cons w = 0.04 | Time w = 0.2 | Total Score | Decision ≥8 |
|---|---|---|---|---|---|---|---|---|---|---|---|---|
| 1 | Farmland location | Open Street Map | 2.52 | 1.76 | 1.53 | 1.04 | 0.81 | 0.48 | 0.36 | 0.18 | 8.68 | accepted |
| | | Google Map | 2.52 | 1.98 | 1.53 | 1.17 | 0.81 | 0.54 | 0.36 | 0.18 | 9.09 | accepted |
| 2 | Weather forecast | Forecast—OWN | 2.24 | 1.76 | 1.53 | 1.17 | 0.81 | 0.54 | 0.32 | 0.18 | 8.55 | accepted |
| | | BMKG | 2.52 | 1.54 | 1.53 | 1.04 | 0.81 | 0.42 | 0.32 | 0.16 | 8.34 | accepted |
| | | World Bank | 2.24 | 1.76 | 1.53 | 1.17 | 0.81 | 0.42 | 0.36 | 0.16 | 8.45 | accepted |
| 3 | Agricultural Financing | Ministry Coordinator of Finance | 2.52 | 1.98 | 1.53 | 0.91 | 0.81 | 0.54 | 0.28 | 0.18 | 8.75 | accepted |
| | | News—Google search | 2.52 | 1.98 | 1.53 | 0.91 | 0.81 | 0.54 | 0.28 | 0.18 | 8.75 | accepted |
| 4 | Land Preparation | IAARD | 2.52 | 1.98 | 1.53 | 0.91 | 0.72 | 0.54 | 0.32 | 0.18 | 8.70 | accepted |
| | | Youtube | 2.52 | 1.98 | 1.53 | 0.78 | 0.81 | 0.48 | 0.32 | 0.16 | 8.58 | accepted |
| | | Cyber extension | 2.52 | 1.98 | 1.53 | 0.91 | 0.81 | 0.54 | 0.32 | 0.16 | 8.77 | accepted |

**Table A3.** *Cont.*

| No. | Functionality | Candidate External Data Source | Access w = 0.28 | Lice w = 0.22 | Sour w = 0.17 | Conn w = 0.13 | Accu w = 0.09 | Comp w = 0.06 | Cons w = 0.04 | Time w = 0.2 | Total Score | Decision ≥8 |
|-----|---------------|-------------------------------|------|------|------|------|------|------|------|------|------|------|
| 5 | Cultivation technology | IAARD | 2.52 | 1.98 | 1.53 | 0.91 | 0.81 | 0.48 | 0.32 | 0.16 | 8.71 | accepted |
| | | Youtube | 2.24 | 1.98 | 1.36 | 1.17 | 0.72 | 0.54 | 0.32 | 0.18 | 8.51 | accepted |
| | | Cyber extension | 2.52 | 1.98 | 1.53 | 0.91 | 0.81 | 0.54 | 0.32 | 0.16 | 8.77 | accepted |
| | | Repositori publikasi | 2.52 | 1.98 | 1.53 | 0.91 | 0.81 | 0.54 | 0.32 | 0.16 | 8.77 | accepted |
| | | Sipindo | 1.40 | 1.10 | 1.36 | 0.65 | 0.81 | 0.42 | 0.20 | 0.16 | 6.10 | rejected |
| | | Itani | 1.40 | 1.10 | 1.53 | 0.65 | 0.81 | 0.42 | 0.20 | 0.16 | 6.27 | rejected |
| | | Digitani | 1.40 | 1.10 | 1.53 | 0.65 | 0.81 | 0.42 | 0.20 | 0.16 | 6.27 | rejected |
| 6 | Seed description | PPVT | 1.40 | 1.10 | 1.53 | 0.65 | 0.81 | 0.48 | 0.32 | 0.10 | 6.39 | rejected |
| | | DBVaritas—DG of Horticulture | 2.52 | 1.98 | 1.19 | 0.91 | 0.72 | 0.48 | 0.28 | 0.16 | 8.24 | accepted |
| | | MyAgri | 1.40 | 1.10 | 1.53 | 0.65 | 0.81 | 0.42 | 0.20 | 0.16 | 6.27 | rejected |
| 7 | Handling Pest disease | IAARD | 2.52 | 1.98 | 1.53 | 0.91 | 0.81 | 0.36 | 0.32 | 0.16 | 8.59 | accepted |
| | | OPETE | 2.52 | 1.98 | 1.19 | 0.91 | 0.81 | 0.36 | 0.32 | 0.16 | 8.25 | accepted |
| | | MyAgri | 1.40 | 1.10 | 1.53 | 0.65 | 0.81 | 0.42 | 0.20 | 0.16 | 6.27 | rejected |
| | | Sipindo | 1.40 | 1.10 | 1.36 | 0.65 | 0.81 | 0.42 | 0.20 | 0.16 | 6.10 | rejected |
| | | Expert system—ICHORD | 2.52 | 1.98 | 1.53 | 0.91 | 0.81 | 0.36 | 0.32 | 0.14 | 8.57 | accepted |
| 8 | Market price | Toko tani—BKP | 2.52 | 1.98 | 1.53 | 0.91 | 0.72 | 0.42 | 0.28 | 0.16 | 8.52 | accepted |
| | | PIHPS | 2.52 | 1.98 | 1.53 | 0.91 | 0.72 | 0.42 | 0.28 | 0.16 | 8.52 | accepted |
| | | Shopee | 2.52 | 1.98 | 1.19 | 0.91 | 0.72 | 0.54 | 0.28 | 0.16 | 8.30 | accepted |
| | | Bukalapak | 2.52 | 1.98 | 1.19 | 0.91 | 0.72 | 0.54 | 0.28 | 0.16 | 8.30 | accepted |
| | | Sayurbox | 2.52 | 1.98 | 1.19 | 0.91 | 0.72 | 0.48 | 0.28 | 0.16 | 8.24 | accepted |
| | | Tanihub | 2.52 | 1.98 | 1.19 | 0.91 | 0.72 | 0.42 | 0.28 | 0.16 | 8.18 | accepted |
| | | info pangan jakarta | 2.52 | 1.98 | 1.53 | 0.78 | 0.63 | 0.42 | 0.24 | 0.16 | 8.26 | accepted |

Note: Access: Accessibility; Lice: License; Sour: Source reliability; Conn: Connectedness; Accu: Accuracy Comp: Completeness Cons: Format Consistency; Time: Timeless.

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
