# Peer review of "A Conceptual Model for Development of Small Farm Management Information System: A Case of Indonesian Smallholder Chili Farmers"

_agriculture, doi:10.3390/agriculture12060866_

Round 1
Reviewer 1 Report
Dear Authors,
Despite the novelty of the research, it needs corrections.
1.“When designing an FMIS, the needs of smallholder farmers must also be considered”. It would be better to provide the reference here or explain the reasons, if it’s your own statement.
2. You should provide the better explanation of current research state (provide the references from 2015 to 2022 years).
Please analyze the following works:
Split-Match-Merge Method with Multi-matcher Algorithm to Handle Data Interoperability Problems in Small Farm Management Information System https://link.springer.com/chapter/10.1007/978-3-030-79757-7_26
Management information system adoption at the farm level: evidence from the literature https://www.emerald.com/insight/content/doi/10.1108/BFJ-05-2020-0420/full/html
Tummers, J., Kassahun, A. and Tekinerdogan, B. (2019), “Obstacles and features of farm management information systems: a systematic literature review”, Computers and Electronics in Agriculture, Vol. 157, pp. 189-204, doi: 10.1016/j.compag.2018.12.044
Henriyadi Henriyadi, The model of data interoperability in farm management information system, July 2021.
A Cloud-Based Digital Farm Management System for Vegetable Production Process Management and Quality Traceability Feng Yang 1,2 , Kaiyi Wang 2,*, Yanyun Han 2,* and Zhong Qiao 1
Raimo Nikkilä, Ilkka Seilonen, Kari Koskinen, Software architecture for farm management information systems in precision agriculture, Computers and Electronics in Agriculture, Volume 70, Issue 2, 2010, Pages 328-336.
Small Family Farms; A Perspective from Indonesia, Challenges and Investment, December 2020 DOI:10.13140/RG.2.2.29704.03849
3. Row 12 “ However, the use of FMIS to support crop cultivation is currently relatively expensive for smallholder farmers” the citation should be putted here.
4. Please revise the study question. What is the most appropriate FMIS conceptual model that can be developed for smallholder farmers? – the study question is defined imprecisely. How you are going to measure this appropriateness? You should explain this.
5. Row 62 “However, this study could not find relevant studies that use agribusiness subsystems in grouping information” – which study?
6. Row 71 “These discussions, however, have proven to be inadequate as they failed to consider smallholder farmers”. – I think “inadequate is too “strong” definition. May be “restricted”.
7. Row 227 Three groups of information in respondents’ characteristics were included in the analysis, namely respondents’ profile, mobile phone ownership, and willingness to try a new application. – How they were selected? The group willingness could be depended on its profile. Ex, younger people more adaptive to modern technologies, their willingness could be higher.
8. Could you provide the statistics how many small chili farms are in Indonesia and explain the choice of size of research group? Have you studied farms or farmers? If farmers, so 1 farmer from 1 farm? Or for example 2 farmers from 1 farm? – the research should be improved.
9. It was found that 72% of the respondents have a mobile phone, so it means that 28% of them will not use your application (about each third farmer)? How many percent of them will be able to use your application? What is the probability of using it? Have you already added the question in your questionnaire, if the farmers be willing to apply your solution?
10. It will be better to change “the rich picture” on algorithm or ULM use-case diagram.
11. The application layer contain the participation of expert and other actors. So, this software will be not only for small farmers. Could you better explain the actors roles?
12. It is great that you have studied the information which farmers need, but why they are looking for this information? Which decisions they will be able to make? If they just looking for the information, they can use the web pages. But how your system will be able to help them in process of decision-making or forecasting? This issue should be better explained.
13. To sum up, in my opinion the description of model should be extended.
14. The statistics and profiles should be better explained. What were the reasons and criteria of choosing the concrete farmers (research sample).
15. Is this just focus study?
16. What is the ratio of the research sample to the number of farmers?
17. The results of research (effects) should be analyzed with the focus on research goal. The research goal should be precised.
18. The literature should be improved, more recent positions are needed.
19. The explanation of AHP method choice should be provided.
20. Have you did the cost or cost-benefit calculations? The costs comparison could be provided.
21. The limitations of this research were discussed, however, they should be named and underlined precisely.
Author Response
Dear Editor and reviewer 1;
Please see the attachment
Best regards
Henriyadi (st118501@ait.ac.th)*
Chutiporn Anutariya (chutiporn@ait.ac.th)*
Vatcharaporn Esichaikul (vatchara@ait.ac.th)*
*Information and Communication Technology Department, School of Engineering and Technology, Asian Institute of Technology, Thailand
https://ait.ac.th/

Reviewer 2 Report
【General comments】
This manuscript entitled “A Conceptual Model for Development of Small Farm Management Information System: A Case of Indonesian Smallholder Chili Farmers” try to find an FMIS that is suitable for small-holder farmers. The topic of this manuscript is meaningful and suitable for Agriculture Journal. Figures in the manuscript is clear. In my opinion, this manuscript needs some revision.
【Specific comments】
Abstract:
l Line 25, delete an “.” in the end of sentence.
Introduction:
l How is the FMIS application in Indonesia? Please provide additional information.
Methodology:
l Line 115, why only choose Chili in your case study?
l Line 131, why only choose the West Java province?
Results:
l Line 232, line 240, line 248, please standardize the format of subheadings, some of them are overstriking, others are not.
l Table 3. What the meaning of “n.a”? Please add related introduction in the end of table
l Table 6 ,7,8, and 9, please standardize the format of these tables as other tables in your manuscript. The upper and lower lines are different in thickness. It’s better to use three-line table in the sci-tech paper.
Discussion:
l The results in your study are interesting and abundant, but the discussion is too simple. Please add more related discussion.
l Only one reference in your discussion part. Please add more references to validate your results and reasoning.
References:
l Most of your references are too old (before 2015). Please refresh references in recently five years.
Author Response
Dear Editor and Reviewer 2;
Please see the attachment.
Best regards
Henriyadi (st118501@ait.ac.th)*
Chutiporn Anutariya (chutiporn@ait.ac.th)*
Vatcharaporn Esichaikul (vatchara@ait.ac.th)*
Information and Communication Technology Department, School of Engineering and Technology, Asian Institute of Technology, Thailand
https://ait.ac.th/

Reviewer 3 Report
Paraphrasing the paragraph of tables 1 and 2, so that they fit with the table data or modifying the table.
Rearrangement the tables

Author Response
Dear Editor and Reviewer 3;
Please see the attachment.
Best regards
Henriyadi (st118501@ait.ac.th)*
Chutiporn Anutariya (chutiporn@ait.ac.th)*
Vatcharaporn Esichaikul (vatchara@ait.ac.th)*
*Information and Communication Technology Department, School of Engineering and Technology, Asian Institute of Technology, Thailand
https://ait.ac.th/

Reviewer 4 Report
In the submitted manuscript, the authors address the need of developing a robust Farm Management Information Systems (FMIS) for smallholder farmers and therefore, suggest a new FMIS applicable to small-scale agricultural farms for effective and efficient management practices. The study is quite interesting, and the findings will be useful for the farmers if disseminated properly. The reviewer expects to see more follow-up studies by this research team.
Author Response
Dear Editor and Reviewer 4
Please see the attachment.
Best regards
Henriyadi (st118501@ait.ac.th)*
Chutiporn Anutariya (chutiporn@ait.ac.th)*
Vatcharaporn Esichaikul (vatchara@ait.ac.th)*
*Information and Communication Technology Department, School of Engineering and Technology, Asian Institute of Technology, Thailand
https://ait.ac.th/

Round 2
Reviewer 1 Report
Dear Authors, in my opinion, although the interesting concept, the sense of creating this technology has still not been clearly described/proven, although the overall quality of the manuscript was improved. I hope it will be done in future articles.
Reviewer 2 Report
All my comments were fully addressed, and I have no additional comment.